# Position: Interpretability Can Be Actionable

**Hadas Orgad** [1]

**Fazl Barez** [* 2 3]   **Tal Haklay** [* 4]   **Isabelle Lee** [* 5]   **Marius Mosbach** [* 6 7]   **Anja Reusch** [* 4]   **Naomi Saphra** [* 1 8]
**Byron C Wallace** [* 9]   **Sarah Wiegreffe** [* 10]   **Eric Wong** [* 11]
**Ian Tenney** [12]   **Mor Geva** [13]

## Abstract

Interpretability aims to explain the behavior of deep neural networks. Despite rapid growth, there is mounting concern that much of this work has not translated into practical impact, raising questions about its relevance and utility. This position paper argues that interpretability *can* be actionable. The central missing ingredient is not new methods, but evaluation criteria: interpretability should be evaluated by *actionability*—the extent to which insights enable concrete decisions and interventions beyond interpretability research itself. We define actionable interpretability along two dimensions—concreteness and validation—and analyze the barriers currently preventing real-world impact. To address these barriers, we identify five domains where interpretability offers unique leverage and present a framework for actionable interpretability with evaluation criteria aligned with practical outcomes. Our goal is not to downplay exploratory research, but to establish actionability as a core objective of interpretability research.

## 1. Introduction

Interpretability research seeks to explain modern machine learning systems, often by studying model representations, specific behaviors or capabilities, or internal mechanisms. In recent years, it has grown into a large and active research

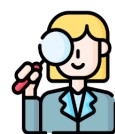

How can interpretability research achieve real-world impact?

Making actionability a core evaluation criterion is required to accelerate progress and achieve measurable real-world impact

**Actionable checklist**

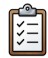

**1. Define a clear goal**
Identify a specific problem that your interpretability question aims to eventually solve.

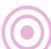

**2. Identify your audience**
Communicate insights according to different stakeholders: developers, practitioners, policymakers.

**3. Propose concrete actions**
Articulate what decisions or interventions your insights enable.

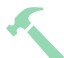

**4. Validate empirically**
Implement the proposed action yourself and demonstrate its effects.

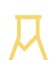

**5. Evaluate in realistic settings**
Apply methods to large-scale models and non-synthetic datasets.

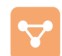

**6. Actionability as success criteria**
- ☐ Surpasses standard baselines (prompting, fine-tuning).
- ☐ Generalizes across setting variations and seeds.
- ☐ Produces targeted effects without degrading other capabilities.
- ☐ Yields useful explanations for target audience.

*Figure 1.* Actionability checklist for interpretability research.

---
[*]Equal contribution   [1]Kempner Institute at Harvard University [2]University of Oxford [3]Martian [4]Technion—IIT [5]University of Southern California [6]Mila – Quebec AI Institute [7]McGill University [8]Boston University [9]Northeastern University [10]University of Maryland [11]University of Pennsylvania [12]Google DeepMind [13]Tel Aviv University. Correspondence to: Hadas Orgad <hadasorgad@fas.harvard.edu>.

*Proceedings of the 43rd International Conference on Machine Learning*, Seoul, South Korea. PMLR 306, 2026. Copyright 2026 by the author(s).

area (Mosbach et al., 2024; Maslej et al., 2025), driven by the intuition that understanding models should help make them more reliable, efficient, safer, and aligned with human values (Bereska & Gavves, 2024).

Despite its growth, interpretability work is often seen as lacking practical impact such as informing changes to models,

training practices, deployment decisions, or policy (Krishnan, 2020; Greenblatt et al., 2023; Potts, 2025), motivating calls to focus on clearly demonstrable outcomes beyond "understanding" itself (Haklay et al., 2025b; Upadhyay & Barez, 2025; Nanda et al., 2025; Barez, 2026). Our framing draws in part on discussions from the ICML 2025 workshop on actionable interpretability, which aimed to foster dialogue on leveraging interpretability insights to drive tangible advancements in AI.

**In this paper, we argue that interpretability research should be evaluated not only by how well it explains models, but by what those explanations enable us to do.** That is, interpretability should be held to a standard of *actionability*. The title's "can" is therefore intentional: it pushes back against the view that interpretability only offers understanding, and reflects our claim that interpretability research can become actionable if it is built and evaluated toward actionable outcomes.

We contend that the field's impact will be strengthened if it explicitly tracks not only what we understand, but what that understanding enables us to do. We do not, however, argue that all interpretability work must immediately yield actionable outcomes, nor that purely exploratory contributions lack value. Indeed, methodological novelty and demonstrated applications are not at odds—grounding findings in real-world actions holds methods to a higher standard, providing evidence that insights reflect genuine model behavior rather than artifacts of a particular analysis. What is missing in interpretability research is not *methods*, but *evaluation criteria*: a shared framework for determining when interpretability research is successful from a practical, decision-oriented perspective. We therefore advance a framework for actionable interpretability: analyzing current limitations, identifying opportunities for impact, and suggesting practical tools to increase actionability. Figure 1 translates our central thesis into concrete steps for researchers.

**Scope.** We consider interpretability in modern machine learning, focusing on deep learning and foundation models. Although we draw many examples from LLMs, our arguments apply broadly to any deep neural networks domain which may require explanation. This is a position paper: rather than an exhaustive survey, we propose actionability as a unifying lens for evaluating interpretability work.

We use interpretability to refer primarily to work that analyzes models to understand their behavior, representations, mechanisms, or internal computations. This overlaps with explainable AI (XAI), but the emphasis differs. XAI often treats explanations as human-facing communication objects, evaluated by how well they help users understand or act on model outputs. By contrast, much recent interpretability work in ML studies models as objects of scientific inquiry, often without specifying a user or downstream decision. Our argument is directed mainly at this latter community.

The paper is organized as follows: Section 2 defines actionable interpretability. Section 3 diagnoses the barriers that currently prevent interpretability from achieving real-world impact. The rest of the paper paves the way towards more actionable interpretability. Section 4 identifies opportunities for actionability. Section 5 presents a framework for categorizing actions and Section 6 discusses evaluation criteria aligned with actionability. Section 7 addresses counter-arguments. Section 8 reviews related work. Section 9 concludes with an actionable checklist for researchers, summarized in Figure 1.

## 2. Defining Actionable Interpretability

We consider a work[1] to be *interpretability-oriented* if it aims to explain or analyze an AI model—for example, works that analyze model representations, explain specific behaviors or capabilities, or discover internal mechanisms.[2] With this distinction in place, we provide the following definition:

**Actionable Interpretability** An interpretability-oriented work is actionable if it produces *insights* about an AI model that inform or guide *actions* toward non-interpretability objectives.

**Insights** are outputs of interpretability work: findings about how models represent or process inputs, explanations of internal mechanisms, or methods that clarify behavior.

**Actions** (toward non-interpretability objectives) are decisions made by humans in response to interpretability insights that would not have been taken otherwise. These fall outside the scope of interpretability itself and ideally lead to concrete improvements such as enhanced performance, better-calibrated trust, or improved safety.

### 2.1. Dimensions of Actionability

In practice, actionability is more fine-grained and not binary. Interpretability-oriented work can support different levels of actionability, which we characterize along two key dimensions: *concreteness* and *validation*.

**Concreteness** captures how precisely an action is articulated. At the low end are vague suggestions ("could inform safety research") or no suggestions at all; at the high end are exact specifications with implementation details.

**Validation** captures empirical support for an action's utility.

---

[1]By "work", we refer broadly to research or engineering contributions, including methods, models, analyses, benchmarks, and empirical studies.

[2]This excludes broader uses of "interpretability" that refer to model-development diagnostics, architecture-search heuristics, or optimization insights rather than explanations of model behavior.

At the low end, actions are untested hypotheses; at the high end, they are systematically evaluated with quantitative or qualitative evidence of meaningful outcomes beyond interpretability research itself.

Together, these dimensions span a space for situating interpretability work (illustrated in Figure 3 in the Appendix):

*Low concreteness, low validation*: Work in this region recommends no specific actions to validate. The insights from this work may, however, inform future work by providing a starting point that others can build upon and test. For example, Geva et al. (2021)'s key-value memory view of MLPs directionally motivated subsequent work on knowledge localization and model editing. Wang et al. (2023), Conmy et al. (2023) and others laid groundwork for circuit-based analysis. While not the emphasis of this paper, such exploratory work is imperative to drive the field forward.

*High concreteness, low validation*: Concrete actions proposed but not empirically validated—e.g., approaches for verifying scientific models to build trust in their predictions (King et al., 2025; Li et al.; Ferreira et al., 2025) or optimizing model deployment and training (Zhao et al., 2025; Chen et al., 2025).

*High concreteness, high validation.* Precise specifications with demonstrated utility, informed by interpretability insights drawn either from the work itself or prior work. Examples include model editing methods leveraging the MLP key-value store view (Meng et al., 2022; Wang et al., 2023; Arad et al., 2024; Fang et al., 2025), are based on sparse-auto-encoders (Gur-Arieh et al., 2025; Ashuach et al., 2025a) or insights into the role of cross-attention layers (Orgad et al., 2023; Gandikota et al., 2024). Representation finetuning (Wu et al., 2024), an alternative to LoRA-based methods, was inspired by interpretability findings. Schut et al. (2025) use concept vectors to uncover novel chess concepts transferable to human players. Anthropic (2025) analyzed internal activations during a safety audit of Claude.

## 3. Why Interpretability Isn't (Yet) Actionable

Despite growing interest, several barriers limit interpretability's real-world impact: misaligned incentives, methodological limitations, and deployment challenges. These reinforce a cycle where actionability is not prioritized, methods lack validation, and deployment yields little feedback. The rest of the paper discusses how to advance actionable interpretability despite these limitations.

### 3.1. Misaligned Incentives

The interpretability community does not sufficiently reward work for demonstrating practical value. Without a strong incentive to prove that interpretability methods deliver real-world value, researchers are less likely to conduct or show interest in actionable interpretability work.

**Publication standards do not require actionability.** Papers can be accepted based purely on methodological novelty, with no requirement to demonstrate applications. Meanwhile, **application-focused work is under-rewarded,** Practical demonstrations may be dismissed as "merely engineering" despite their greater potential impact. We argue that methodological novelty and application demonstration are not at odds—demonstrating applications holds interpretability methods to a higher standard, providing evidence that findings are grounded in reality. This asymmetry—low requirements for actionability combined with low rewards for demonstrating it—substantially reduces researchers' incentive to pursue practical applications.

These issues are not unique to the interpretability field, and also exist in mainstream machine learning (ML) research. However, unlike applied ML, where benchmark performance provides immediate feedback, **interpretability lacks clear signals of success**. Mainstream ML research has a forcing function interpretability lacks: new methods must demonstrate gains on established benchmarks. The field has matured by moving from toy problems to real-world tasks—MNIST to ImageNet, Penn Treebank to diverse downstream tasks. However, the interpretability field has yet to fully mature, lacking agreed-upon standards.

### 3.2. Methodological Limitations

These incentive gaps often manifest as concrete technical problems that prevent interpretability insights from translating into action. In this section, we outline such technical problems and associated methods.

**Lack of actionable insights.** Interpretability work often fails to articulate how findings can inspire concrete actions. This limitation was reflected in the ICML 2025 workshop on Actionable Interpretability, where in 21.8% of the submitted papers, at least one reviewer explicitly flagged the work as insufficiently actionable. Mosbach et al. (2024) showed that although interpretability papers are cited, their impact is predominantly conceptual—most citations do not credit changes to training, architecture, or evaluation. While foundational work may eventually drive actionability (Bau, 2025), the field should explicitly reflect on how insights matter beyond its boundaries.

**Oversimplified setups.** Much research uses simplified tasks and small models. For instance, many mechanistic studies on LLMs focus on single next-token predictions (Mueller et al., 2025), whereas real usage involves multi-token generation. These settings are valuable as controlled testbeds, but their insights may not transfer to realistic settings. Recent work by Haklay et al. (2025a) has begun addressing

these limitations with circuit discovery that handles variable-length inputs.

**Insufficient comparative analysis.** Many works lack rigorous comparisons against alternative approaches and fail to evaluate robustness across architectures, datasets, and tasks. As Casper (2023) argues, weak evaluation hinders progress toward practical tools. Recent benchmarks have begun to address this limitation, highlighting the importance of empirical comparisons. AxBench (Wu et al., 2025) showed that prompting and finetuning often outperform interpretability methods for LLM steering. MIB (Mueller et al., 2025) evaluates both circuit localization and causal variable localization—two widely studied directions that previously lacked a means to compare methods.

### 3.3. Deployment Challenges

Even when interpretability methods offer practical value, several barriers hinder their adoption.

**Technical complexity.** To employ interpretability techniques, a user must deeply understand model internals and be familiar with specialized libraries (Nanda & Bloom, 2022; Fiotto-Kaufman et al., 2025). Those outside the community often lack the expertise required (Ashtari et al., 2023) and so rarely adopt these methods, especially when simpler alternatives exist.

**The open-weights assumption.** Most methods require direct access to weights and activations, restricting applicability to open-weight models. This creates a tension: interpretability is often motivated by safety concerns around powerful frontier models, yet these same models are often proprietary and therefore resistant to such analysis.

## 4. Making Interpretability Actionable

In Section 3, we identified limitations that prevent interpretability research from delivering sustained practical impact. Here, we turn to solutions: we identify opportunities where interpretability is uniquely positioned to drive concrete improvements. We identify five domains in which interpretability offers unique leverage—where there is a fundamental advantage from answering *why* questions about the model. In Sections 5 and 6 we discuss the implementation: a framework for actionable interpretability, and its evaluation.

**Problems scaling does not solve.** The scaling hypothesis—the claim that many capabilities improve predictably with increased model size—has proven remarkably successful. Yet certain failure modes persist or even worsen with scale, including hallucinations, catastrophic forgetting, biases and adversarial brittleness. The persistence of these failures across model scales suggests they are fundamental to our current modeling paradigm rather than due to limited capacity. Interpretability offers a path forward precisely because it can identify why models fail. Standard evaluations detect failures but cannot explain their origins or suggest principled interventions. By contrast, interpretability enables sharper hypotheses about underlying mechanisms and reasoning about potential fixes. Even partial insights can rule out hypotheses and guide the design of solutions.

**Alignment.** As AI systems become more capable, ensuring they behave as intended becomes more critical and more difficult. Alignment today still relies on fine-tuning and data curation rather than understanding-driven interventions, but as AI progresses, verifying that AI goals match human goals will shift from aspiration to necessity. Can we credibly claim a model has no deceptive capabilities without understanding its decision-making? Can we audit for backdoors through black-box testing alone? Since alignment concerns what a model optimizes for, it cannot be fully established without interpretability.

**Surgical interventions.** Retraining a flawed model is expensive and risks introducing other unexpected outcomes. Interpretability enables targeted modifications; identifying components responsible for unwanted behaviors allows surgical fixes while preserving other functionality. Though not yet fully practical, this is among interpretability research's most actionable outcomes—it's efficient and affordable. These techniques can enable post-hoc maintenance: bug fixes, policy updates, and rapid responses to new failure modes.

**Architectural design.** Current improvements emerge largely through trial and error—an inefficient, opaque process where success may not scale or transfer to new domains. Interpretability can transform this paradigm by linking specific design choices (data curation, architecture, optimization) to their effects on model behavior. This approach could accelerate progress by narrowing the space of plausible architecture modifications, reducing both labor and compute required.

**Translation of explanation to meaningful concepts.** The most natural role of interpretability is explaining model behavior, yet translating internal signals into meaningful concepts remains a critical bottleneck. In high-stakes domains like healthcare, a radiologist needs to know if an AI-assisted diagnosis depends on clinically relevant features, not which pixels activate; A developer debugging failures needs specific, legible insights than what current circuit discovery methods provide ("layers 7 and 9 interact together"). Automated methods that translate technical explanations into domain-appropriate, *actionable* concepts could unlock interpretability's core promise. This also includes methods that scale interpretability methods beyond a single input or template into a more natural, diverse setting.

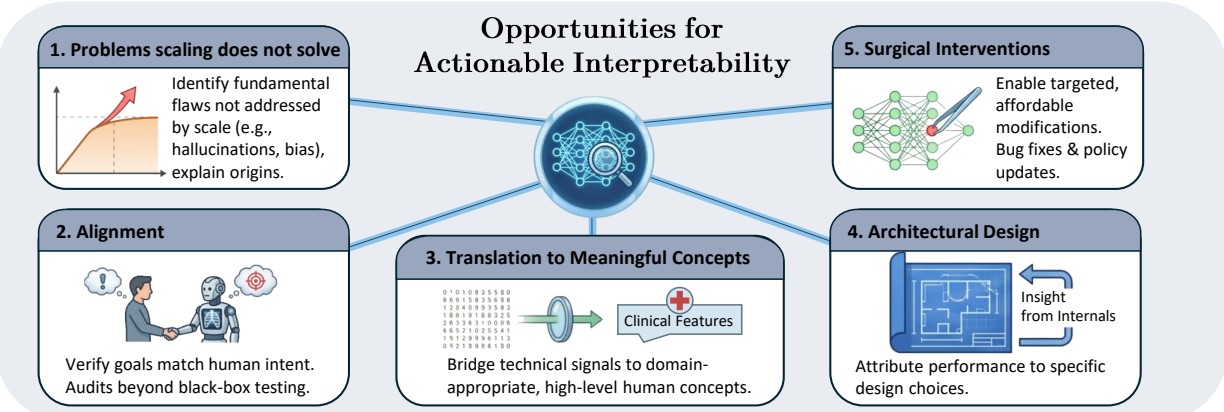

*Figure 2.* Five domains where interpretability offers unique leverage to drive concrete improvements.

## 5. A Framework for Actionable Interpretability

We now present a framework for the actions interpretability enables and the actors who carry them out. This framework is intended to help researchers identify and articulate the actionability potential of their own work. The examples we present throughout this section demonstrate successful cases of actionable interpretability, yet they represent a relatively small fraction of the broader literature.

**Who takes the "action" in "actionable"?** Different stakeholders have different capabilities and motivation, as illustrated in Table 1. **AI developers** may use mechanistic insights to inform model design. **Deployment engineers** may focus on controlling behavior in specific applications. **Domain experts** like clinicians need feature-level rationales to justify diagnoses. **A policymaker** relies on system-level summaries of fairness or compliance. These actors rarely operate in isolation—interpretability outputs should serve as communication interfaces across roles. A clinician's feedback about unreliable explanations may reveal failure modes to engineers. Similarly, a policy maker's compliance requirements may drive developers toward mitigation.

An interpretability work will become more actionable *if it is explicit about its intended audience and the decisions it aims to support.* At the same time, Section 3.3 emphasizes technical complexity as a major barrier to deployment. Taken together, these observations suggest that effective actionable interpretability work must do more than produce insights or analyses: it must specify who can act on its findings, what actions are enabled and how (by providing code or explicit instructions), and where those actions plausibly apply.

We next describe the different types of actions that interpretability insights can drive. We classify them according to what each action affects. Rather than providing an exhaustive taxonomy, we provide representative examples for each.

Additional examples are listed in Appendix B.

### 5.1. Actions that Modify Model Output

Interpretability can inform decisions that directly modify model behavior—changes in training, inputs, weights, or internal computations. These decisions are primarily made by developers and researchers with access to model internals.

**Data curation.** Influence functions can help identify training examples that help or harm model performance. Koh & Liang (2017) used them to detect mislabeled examples and improve accuracy. Han et al. (2020) used them to expose artifacts in training data. More recently, Agia et al. (2025) applied influence functions to robot learning, identifying detrimental demonstrations and achieving state-of-the-art results with only 33% of the original training data.

**Model input.** Interpretability can inform decisions about what inputs to provide to models. Zhou et al. (2024) built on the insight that transformers implement an internal optimizer for in-context learning (Akyürek et al., 2023), using influence functions to identify which in-context demonstrations help versus harm performance.

**Training decisions.** Casper et al. (2024a) built on insights about how models internally represent concepts and used latent adversarial training—perturbing internal representations—to defend against unforeseen vulnerabilities, thereby removing backdoors and improving robustness.

**Direct control.** Interpretability can identify components responsible for specific behaviors, enabling targeted interventions. Model editing modifies weights to insert, remove, or correct behaviors without full retraining (Meng et al., 2022; 2023; Orgad et al., 2023). Runtime interventions steer activations along interpretable directions at inference time (Li et al., 2023; Turner et al., 2023). Concept-based models introduce concepts as intermediate representations, enabling expert-guided control through concept inspection

*Table 1.* Different audiences of interpretability and their actionable outputs.

| Audience | Example Action | Desired Output Type |
|---|---|---|
| **AI developers and researchers** | Curate training data; remove or enhance specific behaviors | Data-point level analysis; behavior modification methods |
| **AI deployment engineers** | Debug application-specific failures | Explanations for model errors |
| **Domain experts and practitioners** | Validate reasoning; refine workflows | Explanations tied to domain features |
| **End users** | Trust or override output, adjust behavior | High-level rationale in human terms, UX/UI for steering model behavior |
| **Policymakers and auditors** | Enforce compliance or transparency | System-level summaries |

and intervention (Koh et al., 2020b; Yuksekgonul et al., 2023; Oikarinen et al., 2023; Steinmann et al.). Explanatory interactive learning methods use feedback on explanations to revise model behavior, for example by penalizing reliance on irrelevant or confounding input regions (Schramowski et al., 2020).

**Safety.** Interpretability can support efforts to remove or suppress unsafe behaviors encoded in model weights. Techniques such as concept erasure (Ravfogel et al., 2020; Elazar et al., 2021) and machine unlearning (Gandikota et al., 2024; Ashuach et al., 2025b; Bourtoule et al., 2021; Cao & Yang, 2015) provide principled approaches for mitigating privacy risks and removing unwanted behaviors by identifying and neutralizing specific learned associations.

## 5.2. Actions about Deployment and Use

Interpretability can inform decisions that end users—including domain experts (e.g., clinicians) and other practitioners—make when interacting with model outputs. Unlike decisions that modify the model's final output, these actions change *what humans do* with model predictions: when to trust them, when to override them, and how to integrate them into their workflows.

**End user decisions.** Prenosil et al. (2025) developed a neuro-symbolic system combining GPT-4 with rule-based expert systems for clinical data extraction, providing the transparency and auditability that enabled radiologists to confidently use AI while maintaining oversight. Activation patching may reveal when models are (overly) relying on patient demographic information when making clinical predictions (Ahsan et al., 2025). Work on uncertainty estimation from internal representations (Kadavath et al., 2022, inter alia) enables users to detect potential errors and make informed decisions about when to trust model outputs.

In medical applications such as embryo selection and histopathology, explanations tied to clinically meaningful features can help practitioners assess whether model predictions rely on appropriate evidence, supporting decisions

about whether to trust, override, or further inspect model outputs (Afnan et al., 2021; Rymarczyk et al., 2024).

**Deployment decisions.** Interpretability enables predictions about where models will fail. Huang et al. (2025) use internal mechanisms to identify out-of-distribution failures during inference, while Li et al. (2025) use them to predict errors on unseen distributions. Such insights support routing decisions—whether to return a model's answer or escalate to alternative methods. Work that detects model errors based on internal representations (Kadavath et al., 2022, inter alia) can also be used in this context. Chen et al. (2024a) demonstrate the potential value, achieving up to 98% cost reduction while matching GPT-4 performance through uncertainty-based routing.

## 5.3. Shaping Future Practice

Beyond immediate interventions, interpretability may offer insights that inform how the field builds and governs future systems. This has longer-term, broader impact.

**Policy and regulation.** Interpretability requirements are increasingly embedded in regulations. The EU AI Act mandates explainability for high-risk AI systems (ArtificialIntelligenceAct.eu, 2025). GDPR's Article 22 (European Union, 2016) restricts automated decision-making with significant effects and requires safeguards, including the right for human intervention. A central challenge to regulation is verification: whether interpretability enables credible claims about the absence of dangerous mechanisms, even with limited access to proprietary model details. This is currently largely unsolved, even for public or open-weight models.

**Learning from superhuman models.** When models exceed human expertise, interpretability becomes a mechanism not just for trust or safety, but for transferring knowledge from machines back to humans. Schut et al. (2025) show that interpreting AlphaZero's (Silver et al., 2018) strategies surfaces novel chess concepts that can teach human grandmasters—demonstrating interpretability's potential for knowledge transfer from AI to humans.

**Development of future models.** Interpretability can shift model design from trial-and-error toward principled engineering guided by an understanding of the computations performed. For example, induction heads in Transformers (Elhage et al., 2021; Olsson et al., 2022) provided a mechanism for in-context learning that traditional state-space models lacked, directly influencing the design of the Mamba architecture's selective state updates (Gu & Dao, 2024).

## 6. Evaluating Actionability

How do we know if interpretability work is actionable? Current practice often evaluates methods against other interpretability techniques or relies on intuitive notions of "understanding". This is insufficient for actionable interpretability. Here, we require metrics that measure whether insights actually enable better decisions and outcomes. We propose evaluation criteria for insights that can enable each of the three action categories from Section 5. All of these criteria share a common principle: *interacting with the world beyond the field of interpretability* rather than solely comparing methods within the field.

### 6.1. Evaluating Actions that Modify Outputs

**Comparative utility against standard baselines.** A major limitation of current evaluation is "grading on a curve"—comparing interpretability methods only against each other. Instead, performance should be measured against standard, pragmatic ML baselines, such as prompting or fine-tuning, and using standard metrics such as accuracy, or benchmark-specific measures. For example, does steering with SAEs improve refusal behavior more than targeted prompting or small LoRA (Hu et al., 2022) adapters? This defines actionability as marginal leverage gained over simpler methods that do not require deep mechanistic understanding.

**Mechanistic faithfulness.** This measures whether an explanation correctly identifies model components causally involved in a specific computation. The broader XAI literature has long studied related notions under faithfulness, fidelity, and causal explanation, including causal accounts of necessity and sufficiency, logic-based explanation, and faithfulness criteria for self-explaining and deep models (Pearl, 2009; Jacovi & Goldberg, 2020; Beckers, 2022; Marques-Silva, 2024; Lyu et al., 2024; Azzolin et al., 2025) In our setting, we focus on a mechanistic form of faithfulness: explanations should identify internal model components whose intervention produces the predicted behavioral change. Evaluation uses intervention-based verification on tasks with well-defined semantics: an explanation is faithful if intervening on identified components produces predicted changes—altering target computations while leaving unrelated behaviors intact. For example, when reverse engineering an LLM's sorting algorithm, one can identify the comparison component and intervene to reliably swap two specific items in the output.

**Generalization.** To address whether an insight holds beyond a specific setting, our metrics must evaluate whether it generalizes, e.g., across seeds, input perturbations, architectures, and scales, without requiring rediscovery. Concrete evaluations may include transferring a discovery method for circuits, features, or mechanisms across models of different sizes, or from toy settings to frontier models. A successful transfer, enabling actionable insights in each setting, indicates that the method captures robust, reusable structure.

**Specificity.** Next, we consider whether an interpretability claim identifies a component that is specifically linked to a distinctive target, rather than a broad correlation. This is evaluated in two ways. First, does the proposed component explain the behavior better than alternatives? This establishes that the finding is genuinely informative rather than arbitrary. Second, when intervening on the component to modify target behavior, do unrelated behaviors remain unchanged? This should be evaluated on standard benchmarks that quantify model capabilities. For example, if a neuron *specifically* controls review sentiment, an intervention may affect the tone while preserving the factual content and performance on unrelated tasks. Interventions that reveal broad effects suggest the component plays a generalized, entangled role in model behavior.

### 6.2. Evaluating Actions about Deployment

**Task-enhancement.** The most direct user-facing metric is whether explanations improve performance on the task the model supports—not the model itself, but human decision-making, speed, or reliance on outputs. For many action types in Section 5, empirical validation can rely on standard automated benchmarks—for example, performance improvement on an automatically evaluated metric.

However, other actions typically requires human-subject evaluations, which are critical since prior work suggests explanations do not reliably improve performance (Spillner et al., 2025). Such evaluations are difficult to conduct well. They require specifying the target user population, designing tasks that reflect realistic decisions, and measuring whether explanations improve the intended human decision process rather than only whether users find them plausible.

**Understandability.** Incomprehensible explanations are unlikely to influence user decisions, even if technically correct. This is especially pronounced in high-stakes settings where practitioners face severe penalties for errors. For example, in medicine, clinical training requirements and legal liability create high barriers to AI adoption, even for superhuman systems. Importantly, understandability is orthogonal to faithfulness—an explanation may accurately reflect model

behavior while failing to be usable. We expand on these evaluations in Appendix C.1.

**Reliability.** Even when explanations improve task performance and are understandable, they may fail to be actionable if perceived as unreliable. Explanations that vary substantially across random seeds or minor perturbations introduce uncertainty that undermines trust. While related to generalization (Cf. Section 6.1), reliability focuses on within-task stability—whether a user can expect consistent explanations across repeated or slightly varied contexts. This framing is especially important in high-stakes domains where explanations guide interventions and brittle explanations are often viewed as unsafe or uninformative (Ghassemi et al., 2021; Arun et al., 2021). We provide examples for measuring reliability in Appendix C.2.

### 6.3. Evaluating Actions Shaping Future Practices

In policy contexts, interpretability acts as an institutional lever rather than a scientific diagnostic. Actionability should be measured by whether methods enable feasible AI governance by regulators and safety teams (Upadhyay & Barez, 2025), and not by depth of insight for a handful of researchers staring at neuron visualizations. From this perspective, interpretability is policy-actionable to the extent that it expands feasible governance interventions: supporting safety audits, interpretable proxy models, or verifying the absence of dangerous mechanisms. Practically, interpretability should reduce monitoring and mitigation costs relative to blunt instruments like pausing deployment, supporting concrete policy tools (e.g., risk audits, model cards, licensing regimes) while remaining legible to non-experts.

## 7. Alternative Views

**Is actionability the right goal?** Some defend interpretability as basic science regardless of actionability. Bau (2025) argues for curiosity-driven research since we do not yet know which techniques may permit actionable insights. We do not disagree. A narrow interpretation of actionability as a strict requirement could shift incentives toward short-term, application-oriented work at the expense of basic research, which is not our intent. Basic and actionable interpretability should be viewed as complementary rather than competing: basic research creates the foundation that enables actionability later, even when the connection is not immediate. What we advocate for is not that all work must be actionable, but that all work should reflect on where it sits relative to actionability: what downstream problem it may eventually contribute to, and what future steps might connect its findings to concrete outcomes, even if speculative. This is a low bar that does not penalize basic research but creates a norm of reflection that increases the chance findings will eventually become actionable.

**How should actionability be defined and measured?** Many believe interpretability's value lies in AI safety (Nanda, 2022; Olah, 2023; Anwar et al., 2024; Amodei, 2025; Marks et al., 2025; Shah et al., 2025). Some argue safety is the *singular* actionable goal (Greenblatt et al., 2023; Nanda, 2022; Nanda et al., 2025; Hendrycks & Hiscott, 2025; Marks, 2025), and performance improvements represent "dual use" problem (Segerie, 2023; So8res, 2023; Shovelain & Mckernon, 2023). We argue for a broader framework centered on human users, encompassing both safety and performance improvements.

**Is actionability achievable?** For practitioners whose priority is building better models, there must be decisive evidence that interpretability methods outperform alternatives with minimal additional effort. We currently lack this evidence for many research lines, contributing to distrust within the broader ML community. However, we can reduce skepticism by ensuring baselines include non-interpretability methods, as we discuss in Section 6. Recent community efforts aim to unify the discussion around actionability (Haklay et al., 2025b). Many remain optimistic, and the community is actively pivoting (Gao, 2025; Ho, 2025; Steinhardt & Schwettmann, 2024; Marks, 2025). We have additionally laid out arguments in this paper for why interpretability *is already actionable* in many scenarios (Section 5).

It would be premature to discount actionable interpretability when the field is still at an early-stage compared to to other scientific disciplines, and our objects of study have only emerged in their current form in the past five years.

## 8. Previous Work

**Previous conceptual and position work.** Lipton (2018) pointed out that interpretability is an overloaded term, and distinguishes between *transparency* (understanding how a model works) and *post-hoc interpretability* (explaining its decisions after the fact). Miller (2019) argues that interpretability requires attention to user and social context because much work neglects decades of findings from from philosophy, psychology, and cognitive science research which highlight how explanations should be grounded. Similarly, Jacovi & Goldberg (2021) emphasize the role of *social attribution* in explanations, namely the implicit attribution of intent to models. Rudin (2019) advances this perspective, suggesting that researchers should abandon post-hoc explanations of models entirely and instead focus on inherently interpretable models.

Others (Doshi-Velez & Kim, 2017) argue that without clear criteria, interpretability research may prioritize intuitively appealing methods over practically valuable ones. While their emphasis aligns with actionable interpretability, we contend evaluation should focus on the specific interventions

and decisions the insights enable. Calderon & Reichart (2025) note that NLP interpretability often fail to generalize beyond their initial domains and stressed the importance of defining stakeholders. While complementing to our view, our focus is on translating insights into actionable outcomes.

Most recently, Nanda et al. (2025) advocate a "pragmatic" approach to interpretability that focuses on solving specific problems rather than solely reverse-engineering models, using meaningful "proxy tasks" to drive rapid iteration. On the other hand, Bau (2025) argues for the importance of curiosity-driven research, noting that we cannot yet predict which interpretability techniques may yield actionable insights in the future.

More broadly, the XAI literature has developed extensive taxonomies of explanation goals, techniques, and application domains (Došilović et al., 2018; Mersha et al., 2024). These perspectives are closely related to ours, but differ in emphasis: XAI often treats explanations as communication objects intended for human users, whereas much recent interpretability work in ML venues studies models as objects of scientific inquiry, focusing on internal computations, representations, mechanisms, and features. Our argument is directed primarily at this latter community, where audience, downstream use, and practical decision-making are not yet standard evaluation criteria.

**Actionable Explainability.** The field of actionable explainability originates primarily from human-AI interaction (HCI) and algorithmic recourse research, focusing on enabling individuals to act on model outputs. An explanation is considered "actionable" if it helps a person understand the changes needed to receive a different outcome in the future (Joshi et al., 2019; Ustun et al., 2019; Karimi et al., 2021; Singh et al., 2024). For instance, increasing income to obtain a loan approval. Other approaches (Singh et al., 2023; Poyiadzi et al., 2020) define actionability as the ability to translate explanations into feasible behavioral changes, while the idea was also extended to human-in-the-loop settings (Saranti et al., 2022), allowing domain experts to directly adjust model parameters.

Actionability is also central to explanatory debugging and interactive machine learning. Prior work shows that explanations can help users diagnose and correct model behavior through iterative feedback (Kulesza et al., 2015) and support interactive learning systems more broadly (Teso et al., 2023). These works anticipate our argument that explanations are most valuable when they enable concrete interventions, but they typically assume tight human-model feedback loops, whereas many modern deep learning systems are large, opaque, and difficult to update or inspect. Our framing of actionable interpretability targets this harder regime. While actionable explainability centers on enabling human action, actionable interpretability reframes interpretability as a way

to also drive concrete improvements in model performance and reliability, not only human understanding.

## 9. Conclusion

In this position paper, we argue that interpretability can have greater real-world impact if actionability is incorporated as a core evaluation criterion. This is not to say that conceptual or theoretical work without immediate practical utility has no place in the field; such research remains valuable and necessary. Rather, making actionability a common evaluation criterion and explicitly tracking what insights make possible can accelerate progress for exploratory work.

We conclude by offering an actionable checklist for interpretability researchers.

1. **Define a clear goal.** Identify a specific problem that your interpretability question aims to eventually solve.

2. **Identify your audience.** Although academic papers are primarily read by researchers, their insights may be acted upon by different stakeholders (e.g., developers, practitioners, policymakers), each of whom may require different framing, language, or levels of abstraction.

3. **Propose concrete actions.** Articulate what decisions or interventions your insights enable.

4. **Validate empirically.** Where possible, implement the proposed action yourself and demonstrate its effects.

5. **Evaluate in realistic settings.** Apply your methods to realistic scenarios, including large-scale models and non-synthetic datasets.

6. **Use actionable metrics**, as described in Section 6. Especially, ask whether your contribution:

   - Surpasses standard baselines (e.g., prompting, fine-tuning) on target metrics.
   - Generalizes across models and other variations of the setting.
   - Produces targeted effects without degrading unrelated capabilities.
   - Yields explanations that are useful for the target audience—and if not, whether they can be translated into a more accessible form.

The burden now falls on the research community: to reward actionable contributions alongside explanatory depth, to establish evaluation criteria that track the utility of interpretability insights, and to build infrastructure that connects understanding to impact.

## Acknowledgments

This work has been made possible in part by a gift from the Chan Zuckerberg Initiative Foundation to establish the Kempner Institute for the Study of Natural and Artificial Intelligence at Harvard University. A.R. was funded through the Azrieli international postdoctoral fellowship and the Ali Kaufman postdoctoral fellowship. B.W. is supported by a grant from Coefficient Giving and the National Science Foundation (NSF), RI 2211954. I.L. and N.S. are supported by a Technical AI Safety Research Grant from Coefficient Giving via Berkeley Existential Risk Initiative. M.M. is supported by the Mila P2v5 grant and the Mila-Samsung grant. E.W. is supported by a grant from the National Science Foundation (NSF), CCF 2442421 and ARPA-H program on Safe and Explainable AI under the award D24AC00253-00.

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

# A. Visualizing the space of Actionable Interpetability work

Figure 3 demonstrates the different types of actionable interpretability work as spanned by the dimensions of actionability.

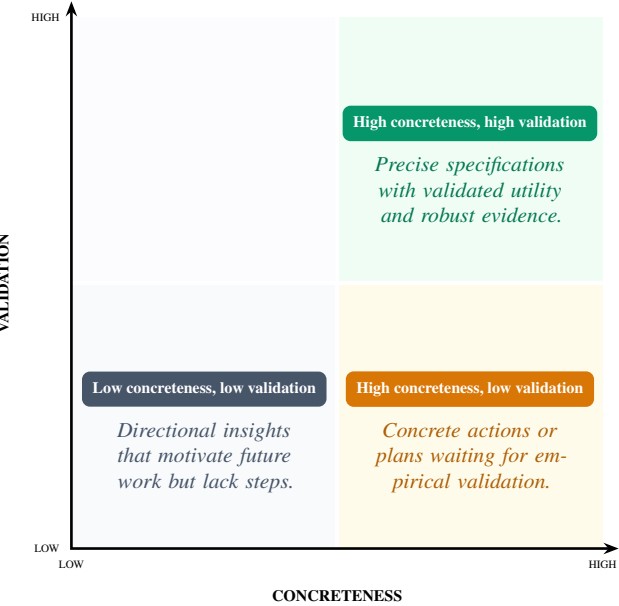

*Figure 3.* The Space of Actionable Interpretability Work.

# B. Additional Examples of Actionable Work

## B.1. Actions That Modify Model's Output

This appendix provides additional examples of actionable interpretability work that modifies model's output.

**Data curation.** Pruthi et al. (2020) estimate the training data influence by tracing how the loss on a test example changes due to each training example during gradient descent. Mangla et al. (2020) use saliency maps to guide adversarial training. They demonstrated that they can identify mislabeled training examples and data poisoning attacks, and that removing the most negatively influential training examples improves model performance. Magnusson et al. (2025) show that strategic data filtering enables small-scale evaluations to forecast large-scale benchmark performance at orders-of-magnitude lower compute cost.

**Model input.** Peysakhovich & Lerer (2023) used attention analysis to discover that models attend more to relevant documents even when not using them, then developed "attention sorting"—reordering documents at inference time to improve retrieval-augmented generation performance.

**Training decisions.** Newman et al. (2025) build on findings that models' internal representations of truthfulness can contradict their outputs (Liu et al., 2023; Orgad et al., 2025). They use these internal representations to curate post-training data, selecting the model's own generations that align with its internal signals of truthfulness, and demonstrate that this approach reduces hallucinations.

**Self-explaining models.** Models that generate explanations as an integral part of their prediction process offer a distinct form of actionability: users can inspect and potentially intervene on the intermediate reasoning. This paradigm includes self-attribution architectures (Agarwal et al., 2021; Brendel & Bethge, 2019; Jain et al., 2020), interpretability wrappers for foundation models (You et al., 2025a), selecting prototypes (Ma et al., 2024; Wen et al., 2024), predicting directly off of concepts (Koh et al., 2020a; Yang et al., 2023; Lai et al., 2024; Yang et al., 2024), or generating programs as explanations that calculate the outcome (Lyu et al., 2023).

**B.2. Actions About Deployment and Use**

**End user decisions.** Recent work demonstrates how the internal representations of LLMs can provide uncertainty estimations about their outputs, enabling users to detect errors and make informed decisions (Kadavath et al., 2022; Azaria & Mitchell, 2023; Gottesman & Geva, 2024; Orgad et al., 2025, inter alia). Building on this foundation, Chakraborti et al. (2025) argue that AI systems in high-stakes domains like healthcare must provide personalized uncertainty estimates to support decision-making: when a clinical decision support system indicates high uncertainty, clinicians can choose to override recommendations rather than following potentially unreliable outputs. Obeso et al. (2025) presented a method for real-time identification of hallucinated tokens in long-form generations.

Chen et al. (2024b) developed a dashboard that exposes the model's internal "user model"" in real time; user studies showed participants valued this transparency for identifying biased behavior.

**Deployment decisions.** Although Chen et al. (2024a) use a separate scoring function rather than interpretability techniques directly, it illustrates how uncertainty estimates enable benefits. Casper et al. (2024b) organized a competition to evaluate whether interpretability tools could help humans detect backdoors implanted in ImageNet-scale CNNs using feature synthesis methods inspired by interpretability research. They achieved 49% human detection rates, significantly outperforming dataset-based attribution methods.

**B.3. Shaping Future Practice**

**Learning from superhuman models.** Goodfire's interpretation of Arc Institute's biological foundation model Evo 2 (Gorton et al., 2025) identifies biologically relevant structure in model representations, demonstrating interpretability's potential to guide scientific investigation.

# C. Additional Examples on Evaluating Actionability

## C.1. Evaluating Understandability

In high-stake decisions contexts, interpretability must present model behavior in a form that aligns with users' existing conceptual frameworks in order to be acted upon. Understandability can be evaluated by measuring how well explanations align with a user's domain-specific reasoning. Benchmarks such as Features Interpretable to eXperts (FIX) (Jin et al., 2024) and its textual extension T-FIX (Havaldar et al., 2025) operationalize this notion by assessing whether explanations correspond to established concepts in domains such as astrophysics or medicine (e.g., cosmological structures or clinical scoring systems like SOFA (Vincent et al., 1996)). For non-expert users, understandability is often captured through plausibility metrics (Agarwal et al., 2024), which evaluate whether explanations appear coherent and reasonable given common-sense expectations.

## C.2. Evaluating Reliability

A large body of prior work proposes metrics for evaluating the robustness of explanations, including sensitivity of feature attributions (Alvarez-Melis & Jaakkola, 2018; Yeh et al., 2019; Kindermans et al., 2019), explanation invariance (Crabbé & van der Schaar, 2023), and provable guarantees on explanation behavior (Blanc et al., 2021; Bassan & Katz, 2023).

One example is work on robustness guarantees in the form of *stability certificates* (Xue et al., 2023; Kim et al., 2024). These certificates explicitly quantify how sensitive a model's predictions are to changes implied by an explanation, such as removing or altering explanatory features. More recent work has extended such guarantees to large-scale foundation models (Jin et al., 2025), chain-of-thought explanations (You et al., 2025b), and clinical applications such as Alzheimer's disease (Achara et al., 2025).

