# OpenReview forum: "Position: Interpretability Can Be Actionable"
_ICML.cc/2026/Position_Paper_Track — ICML 2026 Position Paper Track regular_

### Official Review · Reviewer_varj · 2026-03-07

**Significance:** 1
**Argument Clarity:** 1
**Rating:** 4
**Confidence:** 3

**Questions:**

Why "can" instead of "should" in the title?

**Alternative Views Section:**

No

**Compliance With Llm Reviewing Policy A Conservative:**

Affirmed.

**Discussion Potential:**

1

**Final Justification:**

Raise score on the good faith assumption that the authors will implement the required changes in the final manuscript.

**Paper Summary:**

This paper argues that interpretability in AI/ML research should be actionable and formulate a core evaluation criterion for research manuscripts. Specifically, the paper highlights several steps:
1) Defining a goal
2) Audience identification
3) Propose concrete actions
4) Validate empirically
5) Evaluate in realistic settings
6) Actionability as success criteria.

**Position:**

Yes

**Position In Title:**

Yes

**Related Work:**

4

**Strengths And Weaknesses:**

The paper provides a well-written and extremely well-detailed literature review of the topic of interpretability as. One key strength is Figure 2 which describes how interpretability can be utilized in AI/ML research.

However, a key weakness is that it tends to conflate the topics of "interpretability" with "explainable AI" (XAI), i.e., in the first sentence of the introduction "Interpretability research seeks to explain modern machine learning systems." This may be because XAI is a more well mature field with stronger definitions [1, 2]. It should also be noted that there are interpretability papers that do bring in a more real-world impact, however, this paper is not aware of them [3, 4].

The second weakness of the paper is that it reads more like a survey paper where the authors take a bayesian prior view towards the existing literature and where it should go. Further, the style of the writing in the paper is garbled and at times the paper reads like some corporate Human Resources onboarding document rather than a scientific manuscript.

Finally, the paper's conclusion "the burden now falls on the research community: to reward actionable contributions alongside explanatory depth, to establish evaluation criteria that track the utility of interpretability insights, and to build infrastructure that connects understanding to impact." shows a lot of misunderstanding. Modern ML-driven AI is ends or results-based, not means-based. The result, and what it can do, is most important and ML does that better than rules-based AI of the 1970s-2000s, which is why it now dominates. The motivation is what is important and the motivation for the position exposed in this paper is not very strong.


References:

[1] Mersha, Melkamu, et al. "Explainable artificial intelligence: A survey of needs, techniques, applications, and future direction." Neurocomputing 599 (2024): 128111.

[2] F. K. Došilović, M. Brčić and N. Hlupić, "Explainable artificial intelligence: A survey," 2018 41st International Convention on Information and Communication Technology, Electronics and Microelectronics (MIPRO), Opatija, Croatia, 2018, pp.

[3] Ru, Binxin, et al. "Interpretable neural architecture search via bayesian optimisation with weisfeiler-lehman kernels." arXiv preprint arXiv:2006.07556 (2020).

[4] Mills, Keith G., et al. "Qua2sedimo: Quantifiable quantization sensitivity of diffusion models." Proceedings of the AAAI Conference on Artificial Intelligence. Vol. 39. No. 6. 2025.

**Support:**

3

---

> ### Author Rebuttal · Authors · 2026-03-31
>
> We thank the reviewer for their time and for striving to improve our work, and we will incorporate revisions based on the reviewer’s feedback. We note that some of the critiques would benefit from more specific examples to allow us to make targeted improvements, and we are happy to continue the discussion if the reviewer is able to provide them.
>
> ## "[the paper] tends to conflate the topics of "interpretability" with "explainable AI" " + suggested citations
>
> We appreciate the reviewer's engagement with the interpretability/explainability distinction. The opening sentence uses "explain" in its ordinary English sense, not as a technical term borrowed from XAI. We will revise the opening sentence to make the distinction clearer. The XAI community has long centered the end-user, treating explanation as a communication act directed at a specific audience. Interpretability research, by contrast, specifically in the ML community at which this paper is directed, has largely developed as a study of models as objects in themselves.
>
> We provide a concrete definition that scopes our position, which is focused on interpretability, in section 2. We will add the reviewer's suggested XAI surveys (Mersha et al., 2024; Došilović et al., 2018) as part of a discussion to clarify this point.
>
> Ru et al. (2021) and Mills et al. (2025) use "interpretability" and "explainable insights" differently than our paper's Section 2 definition. Ru et al. (2021) links "interpretability" to discovering architectural motifs via neural architecture search, and Mills et al. (2025) uses "explainable insights" to guide mixed-precision quantization. Neither focuses on understanding internal model computations or representations. Discussing these works highlights the field's terminological ambiguity and will help crystallize our point.
>
> ## "reads more like a survey paper… writing is garbled"
>
> We appreciate the feedback, though we find this critique difficult to address without more specific examples.
> Position papers by nature require surveying the landscape to motivate their argument.
> Sections 3 (diagnosing barriers to actionability), 4 (opportunities for actionable interpretability), 7  (engaging with alternative viewpoints), and 9 (Conclusion and actionable checklist) are explicitly argumentative rather than descriptive.
>
> We take presentation concerns seriously. However, without specific examples of unclear or stylistically problematic passages it is difficult for us to make targeted improvements. We invite the reviewer to share concrete pointers so we can address them directly.
>
> ## Finally, the paper's conclusion ... shows a lot of misunderstanding. Modern ML-driven AI is ends or results-based, not means-based. The result, and what it can do, is most important and ML does that better than rules-based AI of the 1970s-2000s,.... The motivation is what is important and the motivation for the position exposed in this paper is not very strong.
>
> The reviewer argues that ML is results-based rather than means-based — and we agree entirely. This is precisely our point. Interpretability research is the area that has historically not been evaluated by results, and our paper calls for exactly the shift the reviewer advocates: evaluating interpretability by what it enables, not only by how well it explains models. Our conclusion is a call for more results-based evaluation of interpretability, not less.
>
> If our conclusion gives the impression that we are calling for means-based rather than results-based evaluation, we are happy to revise it to make our intent clearer. We would welcome further feedback from the reviewer on which specific passages contributed to this reading.
>
> ## Why "can" instead of "should" in the title?
>
> The choice of "can" over "should" is deliberate and carries meaningful rhetorical weight. "Should" would imply that actionability is achievable and argue it is desirable. "Can" does something more: it asserts that actionability is *achievable*, pushing back against a strain of skepticism in the field that interpretability research is inherently too abstract to deliver practical impact. As we discuss in Section 7, whether actionability is even possible is a genuine open question in the community, and our title is a direct answer to that skepticism.
>
> We agree with the reviewer that there is growing community consensus that interpretability *should* be more actionable. But consensus that something is desirable is not the same as a roadmap for achieving it, and a position paper arguing for something which many already agree on would have little value. Our paper's contribution is precisely to move from "should" to "can": arguing that actionability is within reach if the community adopts the evaluation criteria and practical steps we propose.
>
> We think that this discussion is valuable and will incorporate that in the next revision of our paper.
>
> We hope that we addressed your concerns, and are happy to continue the discussion!

---

> > ### Author Rebuttal · Reviewer_varj · 2026-04-01
> >
> > Authors will need to make revisions for final version of manuscript.

---

### Official Review · Reviewer_Avgk · 2026-03-11

**Significance:** 4
**Argument Clarity:** 4
**Rating:** 6
**Confidence:** 5

**Questions:**

Can you also move outside VLMs/LLMs and SAE in the discussion, it is quite limited branch of interpretability? Even those topics are most cutting-edge, there is still a lot of going on in other areas that make real impact.

**Alternative Views Section:**

Yes

**Compliance With Llm Reviewing Policy A Conservative:**

Affirmed.

**Discussion Potential:**

4

**Final Justification:**

Rebuttal addressed all of mine questions well.

**Paper Summary:**

The work proposes a switch of a focus in interpretability research towards actionability. As actionability, the authors understand impact of a interpretability research towards stakeholders of AI models. Those include but not limits to AI developers, regulatory makers or clinicians. Moreover, the work highlights that people seeing actionable research in interpretability, see it as incremental and engineering, rather than impactful resulting in many rejections. The work introduce actinability checklist and defines axis among those: validation and concreteness.  Then there is a discussion how interpretability can be evaluated among actionability access.

**Position:**

Yes

**Position In Title:**

Yes

**Related Work:**

3

**Strengths And Weaknesses:**

The work is of tremendous importance for ICML community, I found it the most important among my batch of paper to be reviewed.

The position is clearly stated, well argumented and there are evidence and rationale behind the thesis. Very well supported.

Definitely, this position will spark a discussion among the community. I will even bid for opening new research venues.

Some of the related works are cited, I missed few more related to other families of interpretability methods than SAE and circuts, such as: [1], [2] and [3]. Where interpretable AI of different kind has impact on embryo selection, machine unlearning technique, and was also used to identify batch effects in the histology slides dataset, respectively.

[1] Afnan, Michael Anis Mihdi, et al. "Interpretable, not black-box, artificial intelligence should be used for embryo selection." Human reproduction open 2021.4 (2021): hoab040.

[2] Choi, Ching Lam, Alexandre Duplessis, and Serge Belongie. "Unlearning-based Neural Interpretations." arXiv preprint arXiv:2410.08069 (2024).

[3] Rymarczyk, Dawid, et al. "Deep learning models capture histological disease activity in Crohn’s disease and ulcerative colitis with high fidelity." Journal of Crohn's and Colitis 18.4 (2024): 604-614.

Line 122 lacks dot at the end of the sentence :)

**Support:**

4

---

> ### Author Rebuttal · Authors · 2026-03-30
>
> We are grateful for the enthusiastic and thoughtful review, and for recognizing the importance of actionable interpretability for the ICML community. Your constructive suggestions will help strengthen and broaden our work.
>
> We address each point below.
>
> ## Missing citations
>
> Thank you for these highly relevant pointers! We will add all three references in the revision. Afnan et al. (2021) and Rymarczyk et al. (2024) are natural fits for Section 5.2, where we discuss deployment and end-user decisions in high-stakes domains — their arguments are compelling real-world instantiations of actionable interpretability for clinicians. Choi et al. (2024) connects to our discussion of machine unlearning in Section 5.1.
>
> ## "Can you also move outside VLMs/LLMs and SAE in the discussion, it is quite limited branch of interpretability? Even those topics are most cutting-edge, there is still a lot of going on in other areas that make real impact."
>
> We agree that the paper's scope should extend well beyond LLMs and SAEs, though we acknowledge this may not be sufficiently visible in the current presentation. The paper already includes examples from text-to-image diffusion models (Orgad et al., 2023; Gandikota et al., 2024), robot learning (Agia et al., 2025), chess and superhuman models (Schut et al., 2025), clinical and medical applications (Prenosil et al., 2025; Ahsan et al., 2025), and geophysical models (King et al., 2025). It will indeed be valuable to refine the discussions in section 4 (Making Interpretability Actionable) and section 6 (Evaluating Actionability) to incorporate these other domains, and will work on it for the next revision.
>
> Thank you again for the insightful comments and for striving to improve our work; we're happy to continue the discussion as needed.

---

> > ### Author Rebuttal · Reviewer_Avgk · 2026-04-02
> >
> > I was already very positive about the work

---

### Official Review · Reviewer_w5oD · 2026-03-12

**Significance:** 3
**Argument Clarity:** 4
**Rating:** 5
**Confidence:** 4

**Questions:**

- In your vision, does comparing with baselines beyond interpretability research imply that interpretable methods should beat those baselines to be considered successful? In other words, should interpretability efforts aim for comparable performance with simpler methods (while gaining explanations), or should they directly aim for state-of-the-art performance?

**Alternative Views Section:**

Yes

**Compliance With Llm Reviewing Policy A Conservative:**

Affirmed.

**Discussion Potential:**

3

**Final Justification:**

I was already in favor of accepting this paper, and the rebuttal provided by the authors helped further clarify some ambiguous points.

**Paper Summary:**

This paper argues that interpretability can have broader impact that goes beyond solely understanding the internals of a model. The authors define actionability as a diverse set of objectives, that span from improving model performance to increasing practitioners' trust in model predictions, and identify it as a core goal of interpretability research, though often overlooked. Practically, the authors advocate for deeper integration of actionability measures in interpretability research, which could help bridge the gap between the field and the broader ML research.

**Position:**

Yes

**Position In Title:**

Yes

**Related Work:**

3

**Strengths And Weaknesses:**

**Strengths:**

- The paper is well-written, and pleasant to follow. Structure is tidy, and logical flow from current issues to possible solutions is clear.
- The position is well motivated, and issues currently preventing actionable interpretability are thoroughly analyzed.
- Providing concrete examples from existing interpretability works helps understanding the actions taxonomy introduced by the authors

**Weaknesses:**

- It is not entirely clear how prioritizing actionability would interact with with basic, theoretical interpretability research. The authors acknowledge that actionability should be a general goal and not a strict requirement to deem a research effort valuable, but adopting this view could still shift incentives towards short-term, application-oriented contributions.

**Support:**

4

---

> ### Author Rebuttal · Authors · 2026-03-30
>
> We are delighted by the positive assessment and glad the paper's clarity, structure, motivation, and concrete examples were found valuable. We will incorporate changes based on the points raised by the reviewer. We address the specific concerns below.
>
> ## "It is not entirely clear how prioritizing actionability would interact with with basic, theoretical interpretability research. The authors acknowledge that actionability should be a general goal and not a strict requirement to deem a research effort valuable, but adopting this view could still shift incentives towards short-term, application-oriented contributions."
>
> We appreciate this nuanced concern, which we also discuss in Section 7. We agree that a narrow interpretation of actionability as a strict requirement could shift incentives toward short-term, application-oriented work at the expense of basic research — and we want to be clear that this is not our intent.
>
> We see basic and actionable interpretability research as complementary rather than competing. Basic research creates the foundation that enables actionability later, even if the connection is not immediate. A concrete example already discussed in our paper is Geva et al. (2021)'s key-value memory view of MLPs: the work itself was not actionable in our sense, but it directionally motivated subsequent work on knowledge localization, which in turn enabled model editing (Meng et al., 2022), and also influenced architectural design decisions. Curiosity-driven research that cannot yet articulate its downstream value may still be essential precisely because we cannot predict in advance which insights will eventually enable real-world interventions. We cite Bau (2025) in Section 7 as a strong articulation of this view, which we endorse.
>
> What we advocate for is not that all work must be actionable, but that all *work should reflect on where it sits relative to actionability*: what problem it eventually aims to contribute to, and what future steps might connect its findings to concrete outcomes (even if speculative). This is a low bar that does not penalize basic research but creates a norm of reflection that increases the chance findings will eventually become actionable. Practically, we suggest that papers situate themselves along the concreteness and validation axes we define in Section 2, even if they currently occupy the low concreteness, low validation quadrant. We think that this discussion and clarification are important, and we will make this recommendation more explicit in the revised paper.
>
> ## "In your vision, does comparing with baselines beyond interpretability research imply that interpretable methods should beat those baselines to be considered successful? In other words, should interpretability efforts aim for comparable performance with simpler methods (while gaining explanations), or should they directly aim for state-of-the-art performance?"
>
> This is an important clarification. Our view depends on how a work frames its contribution.
>
> When a paper presents an intervention as one of its primary contributions — for instance, steering a model via SAE features to reduce sycophancy as a safety method — the actionability of that intervention is what justifies the work. In this case, comparison to non-interpretability baselines (e.g., prompt the model “do not over-agree”, or fine-tuning) is essential, and the method should ideally match or beat them. We do not require state-of-the-art performance from the outset — progress is valuable — but the gap should be measured and reported so the community can track it over time.
>
> When interpretability is itself part of the value — for instance, concept bottleneck models that trade some performance for transparency — a performance gap relative to black-box methods may be acceptable. But comparison to standard baselines remains important: it quantifies exactly how much performance is sacrificed in exchange for interpretability, which is critical information for practitioners deciding whether to adopt the method. Transparency about this trade-off directly increases the chance of adoption and thus the actionability of the finding.
>
> In both cases, the goal of eventually closing the gap with simpler or black-box methods is the right long-term aspiration, even if not achieved immediately.
> We will incorporate this discussion into a revision of the paper.
>
> Thank you again for the helpful comments; we are happy to continue the discussion as needed.

---

> > ### Author Rebuttal · Reviewer_w5oD · 2026-04-03
> >
> > Thank you for the response, it addresses my concerns. I recommend including such clarifications in the final version of the manuscript. Overall, I maintain my positive evaluation of the paper.

---

### Official Review · Reviewer_w4Mz · 2026-03-16

**Significance:** 3
**Argument Clarity:** 4
**Rating:** 5
**Confidence:** 4

**Questions:**

Please feel free to comment on any of the issues I raised.

**Alternative Views Section:**

Yes

**Compliance With Llm Reviewing Policy A Conservative:**

Affirmed.

**Discussion Potential:**

4

**Final Justification:**

This is a good position paper, well argued and with plenty of potential for down-stream research.  See also my reply to the rebuttal.

**Paper Summary:**

In a nutshell, the authors propose that explainability should be evaluated through down-stream applications of machine explanations, then proceed to list and discuss number of such down-stream tasks and what aspects (e.g. user modeling) should be taken into account into the evaluation.

**Position:**

Yes

**Position In Title:**

No

**Related Work:**

2

**Strengths And Weaknesses:**

TL;DR: timely, well constructed position paper with good potential but also some gaps in the related works.

**Title**:

- CON: the title is not really representative of the main position. It should read: "Explainability enables down-stream tasks and can be evaluated through them" -- except better, of course.

**Significance**:

- PRO: explainability (both old school and mechanistic) has always had trouble finding the "right" form of evaluation. This paper suggests a sensible solution.

- CON, minor: the idea that XAI developers/researchers should always keep the specific end-user in mind is well known; I understand why it is part of the proposal, but it is not a novel element, by far.

- PRO: the core idea is that of viewing down-stream tasks as a natural device for evaluation. To the best of my knowledge this is novel and potentially beneficial.

- CON: in the conclusion (which reads a second call for action, really), points 4 and 5 suggest researchers/developers should validate explanations empirically in realistic settings. This is an integral part of the proposal, but also a major roadblock to adoption. It has been known since the inception of XAI that explanations' interpretability should be evaluated empirically with actual users, but almost nobody does that, at least not properly. The main reason is that user studies require an interdisciplinary background (one has to know how to design a proper study in the first place) that most CS/math/physics curricula do not provide. (Now that I think of it, changing curricula should be part of the call to action.) The vast majority of user studies in the XAI literature are small scale and likely biased: they are designed to make (unprepared) reviewers happy. A second reason is that user studies can be expensive and time consuming, and most ML researchers don't want to deal with the extra costs. I am sure the authors are aware of this (given how careful the text is) and I don't expect this paper to change this state of affairs, but I would appreciate if the authors could at least openly discuss these issues.

**Support/Evidence**:

- The text is well crafted, all arguments are well supported by evidence. I could find any major shortcomings.

**Discussion Potential**:

- PRO: the text is very accessible, widening the potential audience.

- PRO: the topic is timely -- it covers both classical and mechanistic XAI.

- PRO: the position is at the same time sensible and leaves a number of points open for discussion and further research.

**Argument clarity**:

- PRO: arguments are generally well presented and solid. I don't have complaints.

- CON: the generalization paragraph in p 6 is somewhat obscure. What does it mean to transfer a circuit to a different model? I have a hard time understanding how this can possibly work -- unless one is willing to transform the identified circuit, which would inevitably introduce ambiguity into the evaluation.

**Related works**:

- PRO, medium: the paper does a fair job at covering the most recent works...

- CON, medium: ... with some glaring blind spots and a general neglect for earlier works. Specifically, explanations have been used for a while as a device for implementing explanatory debugging:

  - Principles of Explanatory Debugging to Personalize Interactive Machine Learning Kulesza et al.; IUI 2015

  - Leveraging Explanations in Interactive Machine Learning: An Overview Teso et al.; Frontiers in AI 2023

Clearly these works hinge on actionability and are both earlier and more general than knowledge editing (a use case mentioned in the paper).

There is also a growing literature on debiasing concepts and inference algorithms learned by neural networks; the authors only mention Casper et al 2024a, which is not exactly representative of this literature. I recommend to have a look at the above overview and to papers on concept-bottleneck models citing it (say, recent works by Stammer and colleagues).

- CON: the notion of faithfulness is not new, by far. There is an ample literature exploring what it means for an explanation to be faithful and how to measure faithfulness properly. See:

  - Causality, Pearl; 2009. (Notions of probability of sufficiency and probability of necessity)

  - Causal explanations and XAI, Beckers; CRL 2022.

  - Logic-based explainability: Past, present and future; Marques-Silva 2024 (and other works by the same group)

  - Reconsidering faithfulness in regular, self-explainable and domain invariant GNNs Azzolin et al.; ICLR 2025 for a recent example

but feel free to search for "XAI faithfulness" or "XAI fidelity" on scholar for an ample selection.  In the paragraph "mechanistic faithfulness" there is no mention of any work on faithfulness at all - the authors merely outline the basic algorithm, as if it were novel. I am sure this was not the intention, but I think this should be fixed.

**Support:**

4

---

> ### Author Rebuttal · Authors · 2026-03-30
>
> Thank you for the careful read and constructive feedback! We are pleased the paper was considered timely, novel, well-constructed, and accessible. This feedback will be incorporated to improve the manuscript to be more clear and effective.
>
> We largely agree with the reviewer about the historical contributions of the XAI community, and highlight that our position is mainly directed at the current interpretability community publishing technical work in ML venues. Additionally, we claim that the spectrum of relevant actions that can be enabled by interpretability insights extends much beyond human evaluation (section 5).
>
> Below, we address each concern and question.
>
> ## "the idea that XAI developers/researchers should always keep the specific end-user in mind is well known"
> We agree that user-centricity is well-established in XAI. However, interpretability research in ML venues has largely developed as a study of models as objects in themselves, independent of who the findings serve or what they enable. Advocating for this shift within the interpretability community is non-trivial, even if self-evident from an XAI perspective. Audience-awareness has not been a norm in this community in the same way. We are not claiming the idea is novel; we are arguing for its adoption in a community that has largely operated without it. We cite Doshi-Velez & Kim (2017) and Calderón & Reichart (2025) in Section 8 to credit prior work raising this issue. This distinction is a valuable point and we will edit the manuscript to reflect it.
>
> ## "points 4 and 5 suggest researchers/developers should validate explanations empirically … user studies can be expensive and time consuming…."
> You're right that in XAI, empirical validation typically requires expensive user studies. However, for most action types we describe in Section 5, empirical validation does not require user study, but can be measured using standard benchmarks. For the deployment and end-user category (Section 5.2), the concern is more directly applicable, and we will expand this discussion in Section 6.2 of the revision to reflect it.
>
> Recent automated benchmarks such as FIX and T-FIX, mentioned in Appendix C.1, are beginning to approximate expert-level evaluation without requiring experts in the loop.
>
> ## "the title is not really representative of the main position"
> We believe that the suggested reformulation does not fully capture our position. The proposed framing reads to us as a characterization of the goals of the XAI community, which focuses on producing human-facing explanations that help users act on model outputs. Our paper is concerned with interpretability: understanding internal model computations, representations, and features. Beyond terminology, the suggested title is descriptive; our position is normative: interpretability *should* be built and evaluated toward actionable outcomes. We argue that actionability should be a core, first-class evaluation criterion that shapes what research gets done and what counts as success. The title "Interpretability Can Be Actionable" pushes back against discussions in the field that interpretability cannot have real value beyond understanding. We will incorporate this discussion in the revision.
>
> ## "the generalization paragraph in p 6 is somewhat obscure"
> We appreciate you pointing out this ambiguity. By "transferring a circuit", we meant that a discovery method should generalize across models — finding actionable mechanisms in each, not necessarily identical ones. The test of generalization is whether the method enables effective interventions across models (e.g., reducing bias by intervening on biased circuits). We will rewrite this sentence to make this explicit.
>
> ## Missing citations on explanatory debugging and faithfulness
> Thank you for these pointers. We will add Kulesza et al. (2015) and Teso et al. (2023) to Section 8, noting that they operate in a qualitatively different regime — the feedback loop they rely on is much harder to close for modern deep learning systems, and it is precisely this harder problem that motivates our paper's framing of actionability.
>
> We will also expand coverage of the concept bottleneck intervention literature, including Schramowski et al. (Nature Machine Intelligence, 2020) and Steinmann, Stammer et al. (ICML 2024), and welcome any additional pointers.
>
> The "mechanistic faithfulness" paragraph was not intended to present faithfulness as novel, but to describe its current application/misapplication in interpretability. We agree the phrasing might suggest outlining the algorithm for the first time, neglecting prior XAI literature, and we will correct this. We will expand the related work to include Pearl, Beckers, Marques-Silva, Azzolin et al., as well as Jacovi & Goldberg (ACL 2020) and Lyu et al. (2024), which defines and surveys faithfulness evaluation criteria specifically for deep learning models.
>
> We hope that we addressed your concerns, and are happy to keep the discussion.

---

> > ### Author Rebuttal · Reviewer_w4Mz · 2026-04-04
> >
> > Thanks for the rebuttal, it helped clarify some essential points. Now that I read your reasoning, I can agree with the title you chose.  I will update my score shortly.

---

### Decision · Program_Chairs · 2026-04-30

**Decision:**

Accept (regular)

**Comment:**

Reviewers unanimously agreed that one of the main strengths of the paper was the writing, which was well structured and very accessible. The arguments for actionability were well-motivated, clearly presented and convincing for reviewers. Reviewers appreciated the comprehensive coverage of related work which was very effectively used throughout the paper, such as to support its arguments and to give examples of actions from the action taxonomy in Section 5. The only minor weakness raised by some reviewers was that some existing work relevant to interpretability was not mentioned, but the authors have agreed to cite them in the paper. Overall, this is an excellent position paper that will generate significant discussion at ICML.